# Curcumin induces mitochondrial dysfunction-associated oxidative DNA damage in ovarian cancer cells

Qi Bao[1,2,3], Zihan Wang[1], Tingting Yang[4], Xiao Su[4,5], Ying Chen[1], Lifen Liu[1], Qicheng Deng[1], Qingyang Liu[1], Changshun Shao[4], Weipei Zhu[1]*

1 Second Affiliated Hospital of Soochow University, Suzhou, Jiangsu, China, 2 Xuzhou Medical University, Xuzhou, Jiangsu, China, 3 Department of Obstetrics and Gynecology, Liyang Peoples Hospital, Liyang, Jiangsu, China, 4 State Key Laboratory of Radiation Medicine and Protection, Institutes for Translational Medicine, Soochow University, Suzhou, Jiangsu, China, 5 Third Affiliated Hospital of Soochow University, Changzhou, Jiangsu, China

* zwp333xx@126.com

## Abstract

Resistance to chemotherapeutic agents is a critical challenge for the clinical management of ovarian cancer. While curcumin has been reported to possess anti-cancer properties, how it exerts its anti-neoplastic effect on ovarian cancer cells remains to be explored. We here characterized the fate of human ovarian cancer cell lines HO8910 and OVCAR3 treated with curcumin. Cell proliferation, cell death, mitochondrial function, oxidative damage and tumor formation in nude mice were examined. Significant inhibition of proliferation and induction of apoptosis were observed in ovarian cells treated with curcumin. The cancer cells exhibit cell cycle arrest at G2/M phase, mitochondrial accumulation, mitochondrial oxidative stress and high level of DNA damage after curcumin treatment. This effect of curcumin is independent of the BRCA mutation status. Curcumin-induced proliferation inhibition and apoptosis were effectively attenuated by the application of antioxidant N-acetylcysteine (NAC), suggesting that curcumin exerts its anti-cancer effect by inflicting oxidative stress. Curcumin applied at 200 mg/kg intraperitoneal infusion daily also inhibited the growth, oxidative damage, and mitochondrial accumulation of tumor xenografts in vivo. Together, the results indicate that curcumin can exert its anti-tumor effect via inducing mitochondrial dysfunction-associated oxidative DNA damage and can be potentially used in combination with other DNA repair-interfering therapeutics, such as PARP inhibitor, in the treatment of ovarian cancer.

## Background

Ovarian cancer is the most fatal malignancy of all gynecological cancers with the majority of cases (approximately 70%) diagnosed at an advanced stage (FIGO classification of stages III and IV) and having a very low 5-year survival rate [1]. Chemotherapy is an important treatment modality for advanced ovarian cancer, but drug resistance and recurrence are constant challenges. Although targeted therapies have been introduced in recent years, the five-year survival rate remains below 70%. Ovarian cancer cells typically acquire and depend

**Data availability statement:** All relevant data are within the paper and its Supporting Information files.

**Funding:** This work was supported by the Xuzhou Medical University (ZX202414 to BQ) and the Second Affiliated Hospital of Soochow University (XKTJ-XK202006 to ZWP).

**Competing interests:** The authors have declared that no competing interests exist.

on a higher antioxidant capacity for their survival. Therefore, many therapeutic approaches that target the antioxidant defense systems of cancer cells are being explored. According to the Catalogue Of Somatic Mutations In Cancer (COSMIC), many ovarian cancer cell lines, including OVCAR3, A2780, SKOV3 and HEY carrying no BRCA mutation, whereas HO8910 carries BRCA1 5382C mutation [2]. HO8910 and the OVCAR3 cell lines were selected for testing their response to curcumin in this study.

Curcumin, a non-flavonoid polyphenolic compound present in the turmeric plant, is widely recognized for its antioxidant and anti-inflammatory properties[3,4]. It has also been explored for their possible benefits in gynecological and reproductive diseases [5–7] . Curcumin was reported to alleviate the deleterious effects of testicular ischaemia and improve sperm chromatin quality in mice [10]. It can mitigate acrylamide-induced ovarian antioxidant disruption and apoptosis in female Balb/c mice [11]. A diet abundant in curcumin can significantly reduce the incidence of some cancers, including ovarian cancer [8,9]. Laboratory studies showed that Curcumin has many benefits for the reproductive system, it has excellent prospects for the treatment of fertility disorders. Curcumin compensates for the deleterious effects of testicular ischaemia and improves sperm chromatin quality in mice [10], mitigates acrylamide-induced ovarian antioxidant disruption and apoptosis in female Balb/c mice [11]. Curcumin possesses antitumor properties in xenograft models [12], effectively restricts the proliferation of cancer cell [13], stymies cell cycle progression [14], and enhances apoptosis [15] via the induction of oxidative stress [16] and DNA damage [17]. While an anticancer effect of curcumin was reported on ovarian cancer, whether or not this anticancer property of curcumin is exerted through inducing mitochondrial dysfunction has not been well defined [15,18,19], and the exact underpinnings of curcumin-induced oxidative stress and DNA damage are still to be thoroughly characterized. Our research may help clarify whether curcumin exerts its anti-tumor effect by inducing mitochondrial stress.

## Materials and methods

### Cell lines and reagents

HO8910 cells were maintained in our laboratory as previously described [20]. OVCAR3 and IOSE80 were kindly provided by Dr. Renjun Pei, Suzhou Institute of Nano-Tech and Nano-Bionics, Chinese Academy of Sciences. All cell lines were examined and authenticated by short tandem repeat profiling. All cell lines were mycoplasma negative and used within 20 passages after arrival. All cell lines were cultured in DMEM-High Glucose (Gibco) supplemented with 10% FBS (Gibco), 100 IU/ml penicillin, and 100 IU/ml streptomycin (NCM Biotech). Cells with 70%-90% of confluence were detached with trypsin-EDTA 0.25% (Gibco) and rinsed with PBS for two times. Cells then were seeded in 6-well plates at a density of $0.1 \times 10^6$ cells/ml in complete DMEM medium.

### Curcumin treatment of cultured cells

Curcumin (S1848, Selleck Chemicals) was dissolved in DMSO at a stock con-centration of 50 mM and applied to cells in culture at different final concentrations.

### Cell viability assay

HO8910, OVCAR3 and IOSE80 cell lines ($1 \times 10^3$) were seeded in a 96-well culture plate. The number of viable cells was assessed using the day before curcumin treatment. Then, the cells were exposed to different doses of curcumin for 24, 48 or 72 h. After treatment, 10 μL CCK8 reagent (C6005, NCM Biotech), was added to each well and incubated for 3 h at 37 °C. At the end of incubation, the media were carefully removed by aspiration. The absorbance of each

well was measured at 450 nm (BioTek, Cytation5, USA). The experiments were performed in quadruplicate. All experiments were repeated at least three times.

## Colony formation

Single-cell suspensions were generated for each cell line and $1 \times 10^3$ of cells were seeded into six-well tissue culture plates. Then, cells were exposed to different doses of curcumin. Colonies were scored after 10-14 days. All experiments were repeated at least three times.

## Cell cycle analysis

Cell cycle analysis was performed as previously described [21]. After being treated with curcumin for the indicated times, the adherent cells were washed once with PBS, trypsinized, and collected by centrifugation at $400 \times$ g for 5 min. The cells ($10^6$ cells per sample) were fixed in 4 ml of cooled 70% ethanol at –20 °C overnight. After centrifugation at $1000 \times$ g for 10 min, cell pellets were incubated with 0.5 ml of PBS containing 100 μg/ml RNase (Invitrogen) and 5 μg/ml propidium iodide (Sigma-Aldrich) at room temperature for 30 min. Cell cycle distribution was analyzed by measuring DNA content using flow cytometry. The experiment was repeated 3 times and each contained at leasthree biological replicates.

## EdU incorporation assay

HO8910 and OVCAR3 cells plated in 6-well plate were treated with curcumin at the indicated concentrations for 48 h, cell proliferation was detected using the incorporation of 5-ethynyl-2′-deoxyuridine (EdU) with the EdU cell proliferation assay kit (Guangzhou RiboBio Co., Ltd. Guangzhou, China). Briefly, the cells were incubated with 50 μM EdU for 2 h before fixation, permeabilization and EdU staining according to the manufacturer's protocol. The proportion of EdU positive cells was determined by flow cytometry (Becton Dickinson, San Jose, CA, USA). The experiment was repeated 3 times and each contained at leasthree biological replicates.

## Tumor graft models

All animal experiments were in full compliance with the guide for the care and use of laboratory animals and the Institutional Animal Care and Use Committee of the Soochow University. OVCAR3 tumors were generated by subcutaneous (s.c.) injection of $5 \times 10^6$ (suspended in 100 μL PBS) in 4-6-week-old nude mice (BALB/c Nude, Shanghai Southern Model Organism Biotechnology Co.). Tumors were grown for several weeks, according to either a survival schedule (endpoint defined by tumor volume) or a fixed time point. Tumor size was measured by calipers, and the volume was calculated according to the formula $(ab^2) \times \pi/6$, with a and b being the tumor's length and width, respectively, in millimeters, maximally allowed to reach 20mm in diameter before sacrifice. Animals were euthanized at experimental termination or when predetermined Institutional Animal Care and Use Committee rodent health endpoints were reached(e.g., 20 percent reduction in pre-experimental body weight). After anaesthesia by intraperitoneal instillation of 1% lidocaine (15 mg/kg) for 5 min. All nude mice were executed by decapitation.

## Western blot analysis

Cultured cells were lysed in lysis buffer supplemented with protease and phosphatase inhibitors. Protein concentrations of the lysates were determined by the BCA protein assay system (Beyotime). Equal amounts of protein were separated by 10% SDS-PAGE, transferred to PVDF

membrane (Millipore, Billerica, MA). Western blotting was performed using the following primary antibodies: anti-CDC25A (06571, Up State, 1:1000), anti-Cyclin B1 (676686, Proteintech, 1:500), anti-Bax (SC526, Santa Cruz, 1:1000), anti-Bcl2 (SC578, Santa Cruz, 1:1000), anti-Slc7a11(12691, CST, 1:1000), anti-Acsl4 (SC271800, Santa Cruz, 1:1000), anti-PINK1(ab23707, abcam, 1:1000), GAPDH (5174S, CST, 1:5000), Beta-Actin HRP (3700T CST, 1:5000). Appropriate horseradish peroxidase (HRP)-conjugated secondary antibody was selected. The experiment was repeated 3 times and each contained at leasthree biological replicates.

## Cell death assay

Necrotic death was detected by staining cultured cells with propidium iodide (PI, 2 μg/mL) (ST511, Beyotime) and Annexin V(A.V, 2 μg/mL) (640941, Biolegend) in the culture medium. Following 30 min incubation, cells were subjected to flow cytometry. The experiment was repeated 3 times and each contained at leasthree biological replicates.

## Immunofluorescence

Immunofluorescence staining was performed, as previously described. In brief, cultured cells were fixed with 4% paraformaldehyde, permeated with PBS containing 0.2% Triton X-100 for 60 min. After 3 washes with PBS, specimens were incubated with primary antibodies overnight. Immunofluorescence was performed using γ-H2AX (9718S, CST, 1:400), cl-cas3 (9661S, CST, 1:400), Tomm20 (ab186735, abcam, 1:400), Ki67 (ab15580, abcam, 1:400). After 3 washes with PBS, specimens were incubated for 60 min with Alexa Fluor–conjugated secondary antibodies (A0453, Beyotime). After 3 washes with PBS, specimens were incubated for 10 min with Hoechst (C1011, Beyotime), washed, and then cover slipped, and observed with Leica TCS SP8 X fluorescence microscope. The experiment was repeated 3 times and each contained at leasthree biological replicates.

## Detection of ROS, mitochondrial mass and mitochondrial membrane potential (MMP)

Cells were then incubated with 5 μM DCFH-DA (S0033s, Beyotime) or 5 μM Mito-SOX (M36008, Thermo Fisher Scientific) or 5 μM MitoPY1 (B5592, APExBIO) or 5 μM MitoTrackerRed (C1035, Beyotime) in HBSS for the detection of ROS, $O_2 \bullet^-$, $H_2O_2$, mitochondrial counts, respectively. MMP was determined by incubating cells with 2 μM JC-1(C2003S, Beyotime) fluorescent probe, which yields green fluorescence at low concentrations while lights up in red when it accumulates in the mitochondria at high concentrations. Twenty minutes after incubation, cells were washed 3 times with HBSS buffer and subject to flow cytometry. The experiment was repeated 3 times and each contained at leasthree biological replicates.

## Lipid peroxidation assay

Cultured cells were incubated for 20 min with 2 μM BODIPY® 581/591 C11 (D3861, Invitrogen). After washing with PBS, cells were washed 3 times with HBSS buffer and subjected to flow cytometry. The experiment was repeated 3 times and each contained at leasthree biological replicates.

## Statistical analysis

Student's-test or one way ANOVA and Tukey's test were used to determine the statistical significance between experimental groups. Difference was considered significant if the *p* value was less than 0.05.

### Ethics statement

Approval of The research protocol by an Institutional Reviewer Board: N/A. **Informed Consent:** N/A. **Registry and Registration No. of the study/trial:** N/A. **Animal Studies:** The study was approved by the Institutional Animal Care and Use Committee of the Soochow University and WIRB: SUDA20210916A05.

## Results

### 1. Curcumin inhibits proliferation of ovarian cancer cells

Human ovarian cancer cell lines HO8910, OVCAR3 and ovarian epithelium IOSE80 exposed to 2-45 µM concentrations of curcumin for 24, 48, and 72 h were measured for their viability via the CCK8 assay. The inhibitory effect of curcumin (Fig 1A). But in IOSE80, an immortalized ovarian epithelial cell line, the IC50 was significantly higher than ovarian cancer cell lines (S1A Fig). We next examined EdU incorporation in HO8910 and OVCAR3 cells 48 h after curcumin treatment. As shown in Fig 1B, curcumin significantly reduced the percentage of EdU positive cells in a dose-dependent manner. As shown in Fig 1C, treatment of ovarian cancer cells with different concentrations of curcumin dose-dependently reduced colony formation of HO8910 and OVCAR3 cells. To determine whether the growth inhibitory effect of curcumin was mediated by cell cycle blockade, we examined the cell cycle distribution of HO8910, OVCAR3 cells with flow cytometry. As shown in Fig 1D, curcumin treatment for 48 h resulted in an increased proportion of cells in the G2/M phase compared to untreated cells. The G2/M phase arrest was accompanied by altered levels of representative proteins involved in G2/M phase progression (Fig 1E), such as dose-dependent reduction of cyclin B1 and CDC25A. These findings indicate that curcumin can arrest ovarian cancer cells at G2/M stage.

### 2. Curcumin induces apoptosis in ovarian cancer cells

During cell culture, we found that curcumin-treated ovarian cancer cells tended to float and the number of floating cells increased with increasing curcumin concentration. We thus measured apoptosis in HO8910 and OVCAR3 cells treated with curcumin by using flow cytometry. The results showed that curcumin treatment for 48 h dose-dependently increased the percentage of apoptotic cells (Fig 2A). In non-malignant IOSE80 cells, the induction of apoptosis was less pronounced than in cancer cells (S1B Fig). Furthermore, as shown in Fig 2B, the levels of cleaved caspase-3 in HO8910 and OVCAR3 cells were increased after treatment with 5-30 µM curcumin for 48 h. We found that cleaved caspase-3 fluorescence intensity was dose-dependently elevated in cancer cells. Furthermore, an imbalance between the proapoptotic protein Bax and the anti-apoptotic protein Bcl2 was observed. The increased Bax to Bcl2 ratio is known to promote the release of cytochrome C from the mitochondria, followed by caspase-3 activation and apoptosis (Fig 2C).

### 3. Curcumin induces oxidative stress and DNA damage in ovarian cancer cells

Many small natural compounds are known to exert their antineoplastic effect by inducing oxidative stress [20]. Curcumin can increase levels of reactive oxygen species (ROS) and induce DNA damage in cancerous human cells [22–24]. TWe therefore measured the level of ROS in ovarian cancer cells treated with curcumin by using DCFH-DA. When treated with 5-30 µM curcumin for 48 h, HO8910 and OVCAR3 cells displayed significantly higher ROS levels than that in the control cells (Fig 3A). But in IOSE80 cells, ROS levels did not increase after

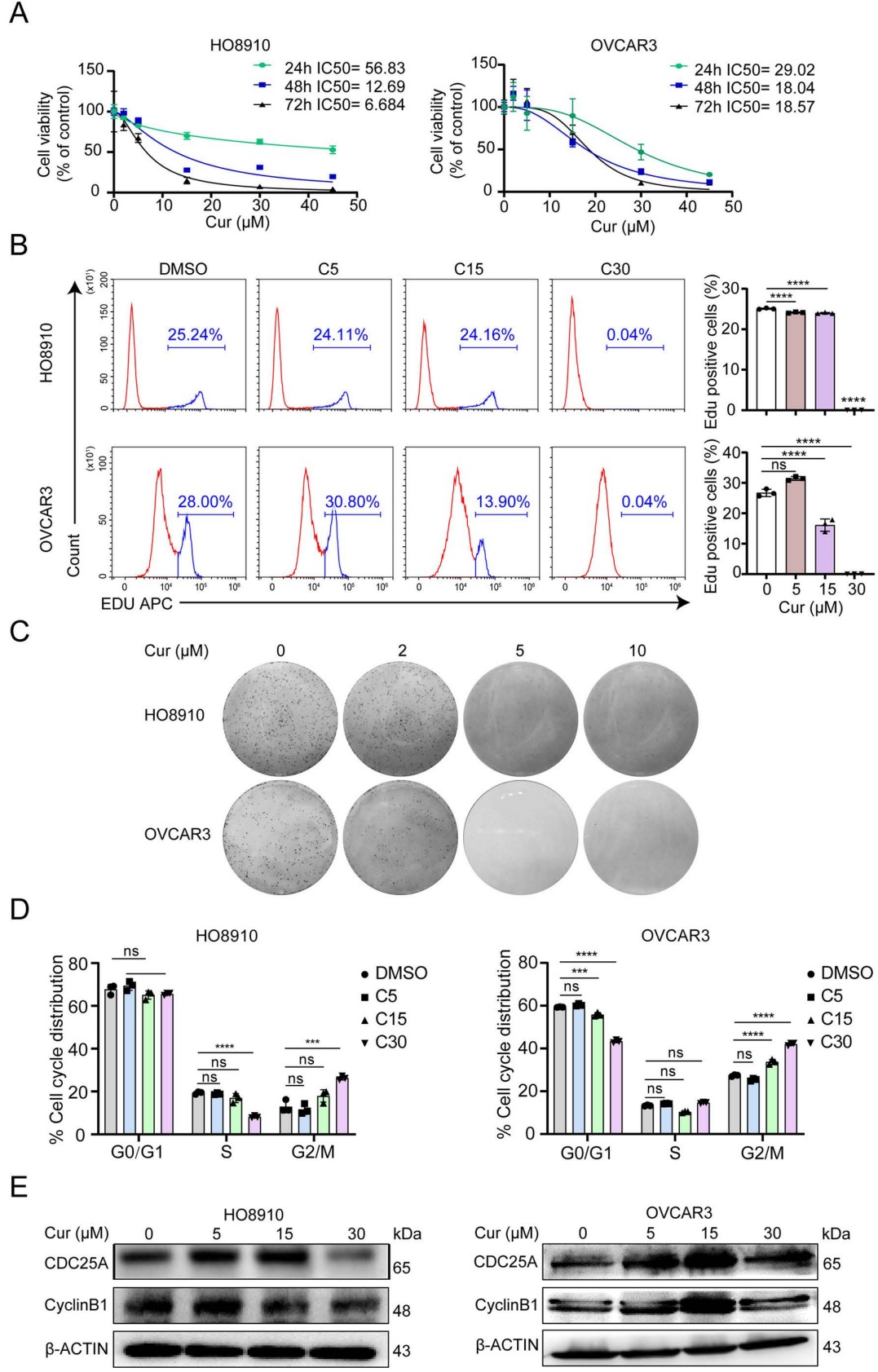

**Fig 1. Curcumin inhibits proliferation of ovarian cancer cells.** (A) HO8910 and OVCAR3 cells were exposed to curcumin (2–45μM) or vehicle control (0.1% DMSO) for 24h,48h and 72h. Cell viability was measured by CCK-8 assay. The

experiments were performed in quadruplicate. (B) EdU proliferation assay was performed 48h after the incubation with 5–30μM curcumin in HO8910 and OVCAR3 cells. Ratio of EdU positive cells is shown as the mean ± S.D. from three independent experiments. * $p < 0.05$, ** $p < 0.01$, *** $p < 0.001$, **** $p < 0.0001$, when compared with control group. (C) Colony formation assay showed inhibitory effect of curcumin in HO8910 and OVCAR3 cells. Colonies were observed until 10-14 days. (D) Effect of curcumin on cell cycle distribution in HO8910 and OVCAR3 cells. HO8910 and OVCAR3 cells were incubated with 5–30μM curcumin for 48h. 0.1% DMSO was used as control. The distribution of cell cycle was assessed by flow cytometry. (E) HO8910 and OVCAR3 cells were exposed to 5–30μM curcumin for 48h. Whole cell lysates were analyzed by immunoblotting with antibodies specific for CDC25A and Cyclin B1. β-actin was used as a loading control.

treatment with 5-30 μM curcumin for 48 h (S1C Fig). Phosphorylated H2AX (also known as γ-H2AX) can be used as a biomarker of DNA double-strand breaks and their repair [25]. We observed that γ-H2AX levels, via immunofluorescence, were remarkably increased in HO8910 and OVCAR3 cell lines following treatment with 5-30 μM curcumin over a 12-18 h period (Fig 3B). In IOSE80 cells, however, γ-H2AX levels did not increase after treatment with 5-30 μM curcumin for 12 h (S1D Fig). Our results indicated a dose-dependent increase in γ-H2AX fluorescence intensity by curcumin in the cancer cells, but not in non-malignant ovarian epithelial cells, indicating that curcumin exerts its cytotoxic effect more preferably in cancer cells.

## 4. Curcumin induces mitochondrial stress in ovarian cancer cells

In the majority of cells, over 90% of oxygen consumption occurs in the mitochondria. Within these organelles, 2% is converted to oxygen radicals in the inner mitochondrial membrane and matrix. Curcumin treatment of ovarian tumor cells caused an increase in ROS levels. We next assessed the level of mitochondrial superoxide anion radical ($O_2•-$) using MitoSox. As shown in Fig 4A, curcumin treatment resulted in a significant elevation in superoxide. We further measured mitochondrial hydrogen peroxide ($H_2O_2$) levels using the fluorescent probe MitoPY1. As shown in Fig 4B, mitochondrial hydrogen peroxide levels were significantly higher in curcumin-treated ovarian cancer cells than those in the control. Curcumin treatment also significantly reduced mitochondrial membrane potential, as measured by JC-1 (Fig 4C). The ATP production was lower in curcumin-treated cells than control cells (Fig 4D). We next examined the mitochondrial mass by performing immunofluorescence of Tomm20, a mitochondrial protein, in HO8910 and OVCAR3 cells treated with 5-30 μM curcumin for 48 h through. As shown in Fig 4E, the fluorescence intensity of Tomm20 was elevated dose-dependently in cancer cells. Staining of MitoTracker Red confirmed the increase in mitochondrial content in cancer cells treated with curcumin (Fig 4F). PINK1 is primarily situated in the mitochondrial inner membrane and is essential for mitophagy, a process that eliminates damaged mitochondria. The expression of PINK1 protein in curcumin-treated cells was analyzed using Western blotting. We found that PINK1 expression was down-regulated in a dose-dependent manner (Fig 4G), indicating that the increase in mitochondria might have been due to decreased mitophagy.

## 5. NAC attenuates the cytotoxic effect of curcumin on ovarian cancer cells

The aforementioned experiments suggest that curcumin may inhibit the proliferation and induce apoptosis of ovarian cancer cells by up-regulating oxidative stress. We next conducted a rescue experiment with the antioxidant NAC to test whether the cytotoxic effect of curcumin on cancer cells was mediated by increased oxidative stress. We treated HO8910 and OVCAR3 cells with NAC (10 mM) or curcumin (5 μM) alone or NAC and curcumin in combination. As shown in Fig 5A, while curcumin reduced the colony formation of cancer cells, the reduction was significantly rescued by NAC. As shown in Fig 5B, the reduced EdU incorporation by curcumin treatment was also significantly rescued by NAC. G2/M phase cell cycle arrest,

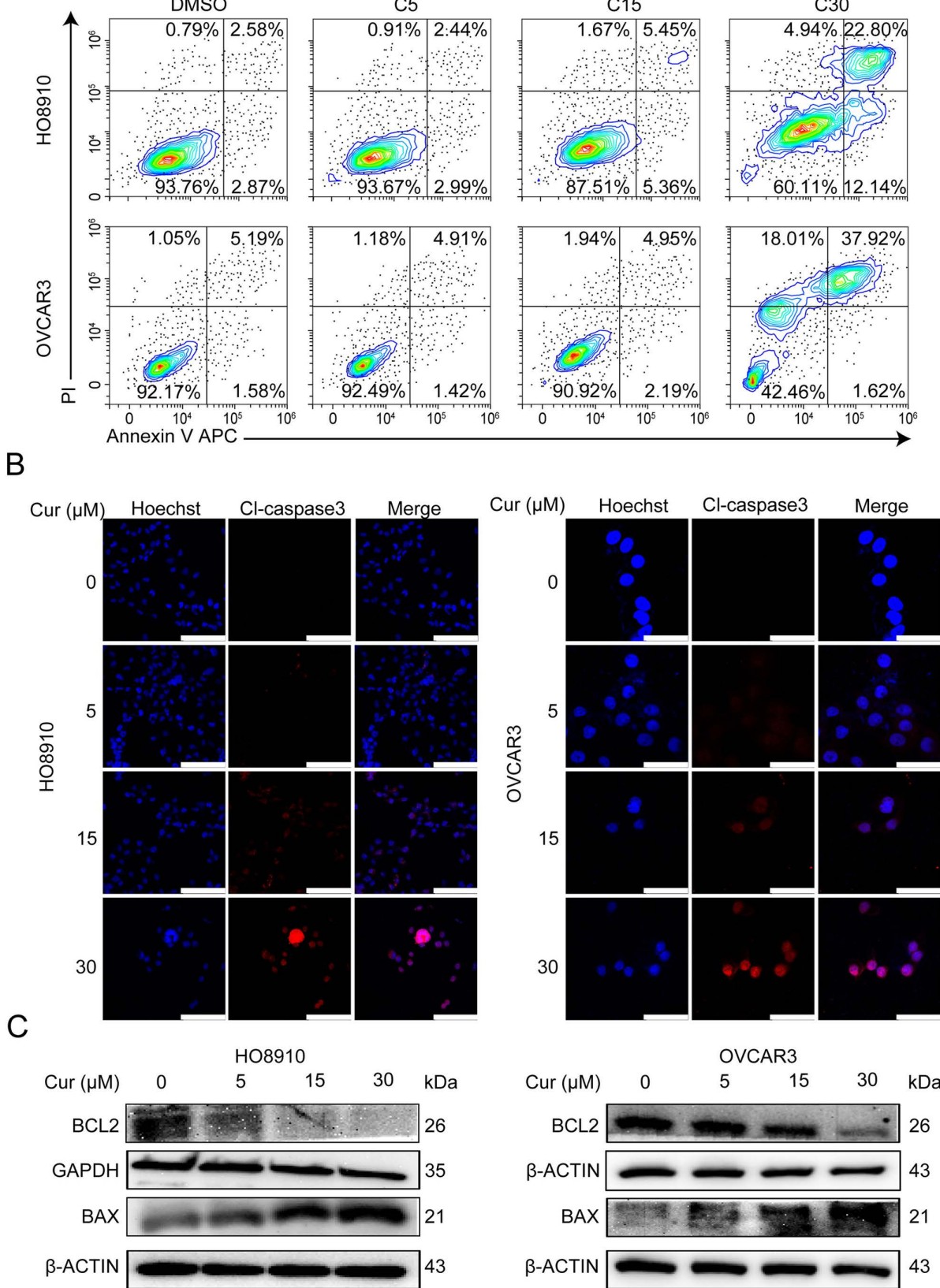

**Fig 2. Curcumin induces apoptosis in ovarian cancer cells.** (A) HO8910 and OVCAR3 cells were exposed to 5, 15, and 30μM curcumin for 48h. 0.1% DMSO was used as control. Cells were processed by flow cytometry using Annexin V and PI staining. The percentage of

Annexin V-positive population indicates early apoptosis induction, and the percentage of Annexin V-negative and PI-positive indicates late apoptosis at every concentration of curcumin. Results shown are representative of three independent experiments. (B) HO8910 and OVCAR3 cells were exposed to 5, 15, and 30μM curcumin for 48h. 0.1% DMSO was used as control. IF images showing the cleaved caspase-3. Hoechst was used for the nuclear staining. Scale bar,96.9 μm and 47.3 μm. (C) HO8910 and OVCAR3 cells were exposed to 5–30μM curcumin for 48h. Whole cell lysates were analyzed by immunoblotting with antibodies specific for Bcl2 and Bax. β-actin was used as a loading control.

apoptosis and decrease in mitochondrial membrane potential induced by curcumin treatment were similarly rescued by NAC (Fig 5C,5D,5E). Likewise, the lipid peroxidation induced by curcumin treatment in ovarian cancer was significantly attenuated by NAC As shown in Fig 5F. These results indicate that the inhibition of proliferation and induction of apoptosis in ovarian cancer cells by curcumin are indeed dependent on the increase in oxidative stress.

## 6. Curcumin inhibits tumor growth in vivo

We further explored whether the tumour-suppressive effects and mechanisms of curcumin in in vivo experiments were consistent with those in vitro. Nude mice bearing subcutaneously injected OVCAR3 cells were randomly divided into 2 groups (n = 5), each mouse was injected once in the left and right axilla (5 million cells), vehicle was used in the control group and curcumin 200 mg/kg via i.p. was used in the treatment group. Tumor volume was measured every three days and growth curves were plotted. After a 15-day treatment, the tumors were extracted and their respective volumes and weights were measured (Fig 6A). The tumor volumes and weights were significantly lower in the mice treated with curcumin than that in control (Fig 6B,6C,6D). There was no difference in the body weight between the two groups (Fig 6E). As shown in Fig 6F, immunofluorescence for Ki67, cleaved-caspase-3 and Tomm20 indicated decreased proliferation, increased apoptosis and elevated accumulation of mitochondria in tumor tissues treated with curcumin.

## Discussion

Chemotherapeutic resistance and recurrence are major challenge in the treatment of ovarian cancer. Conventional chemotherapeutic agents also have tremendous toxicity. Curcumin, a dietary natural compound, was shown to possess anti-tumor effects by inhibiting tumor growth and inducing apoptosis in previous studies, but the mechanisms by which curcumin induces apoptosis in tumor cells are less explored. In this study we observed elevated oxidative stress, increased DNA damage, elevated mitochondrial reactive oxygen species (ROS) levels, decreased ATP production, increased mitochondrial accumulation, and decreased mitochondrial autophagy in curcumin-treated ovarian cancer cells, indicating that curcumin might exert its anti-tumor effect via inducing mitochondrial stress.

Oxidative stress has been implicated in the pathogenesis of ovarian cancer [23,24,26,27]. In response to the accumulation of ROS induced by certain chemotherapies, cancer cells usually produce reducing substances to alleviate oxidative stress, which gradually leads to drug resistance [25]. ROS are produced by the metabolism of oxygen and include superoxide anion ($O_2\bullet-$), hydrogen peroxide ($H_2O_2$), and hydroxyl radical ($\bullet OH$). When produced in excess, they lead to the oxidation of DNA, proteins, lipids and other molecules, and potentially cell death [28]. Cell cycle progression is a biological process regulated by multiple redox-dependent signaling pathways to ensure proper cell division, and the effects of ROS on cell fate oscillate throughout the cell cycle [29] and the effect of ROS on cell fate is variable throughout the entire cell cycle. Intracellular $H_2O_2$ levels increase as the cell cycle progresses, reaching a maximum during the G2/M phase [30]. Interestingly, many cancer therapeutics

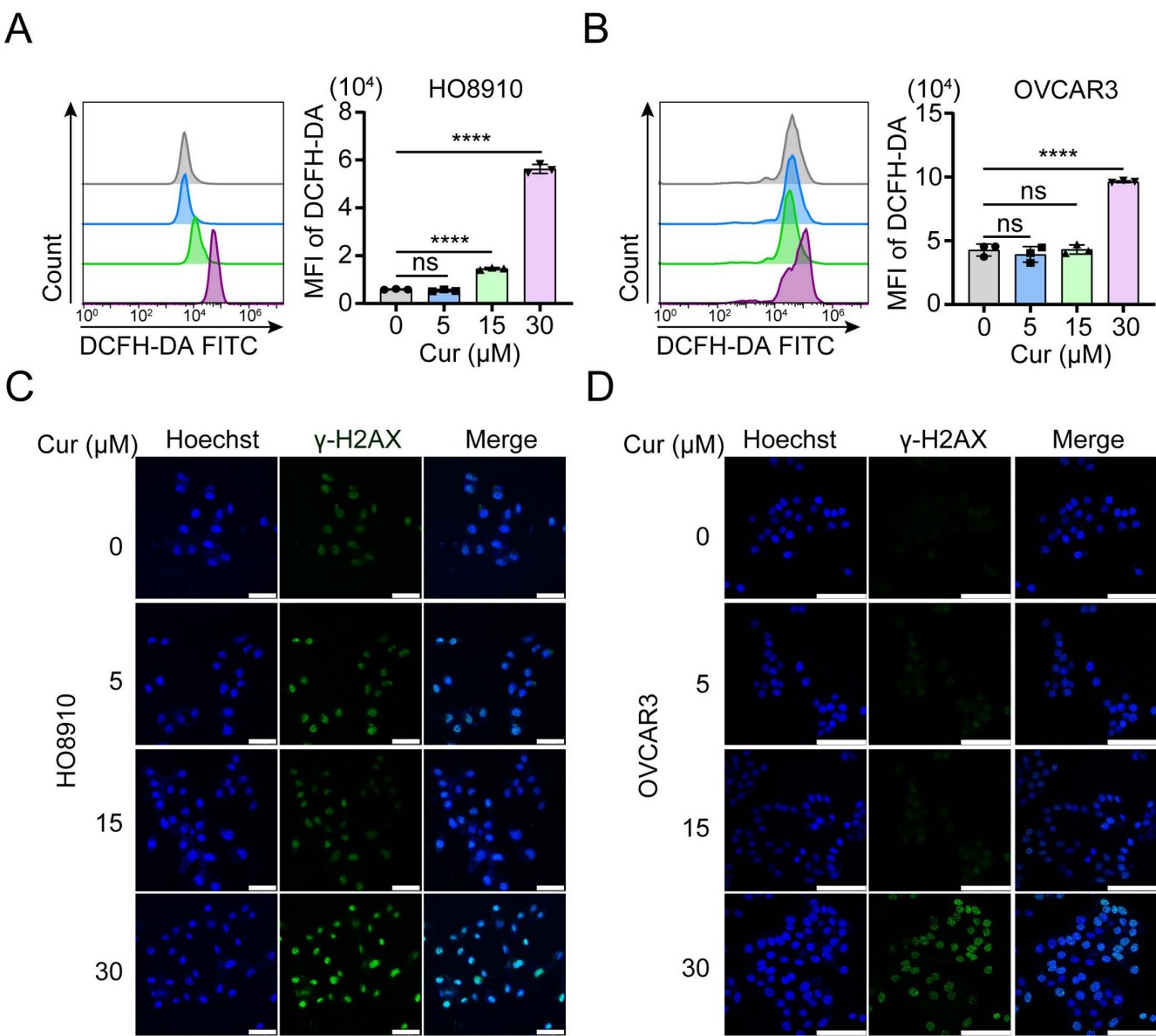

**Fig 3. Curcumin induces oxidative stress and DNA damage in ovarian cancer cells.** HO8910 and OVCAR3 cells were exposed to 5, 15, and 30μM curcumin for 48h. 0.1% DMSO was used as control. $*p < 0.05$, $**p < 0.01$, $***p < 0.001$, $****p < 0.0001$, when compared with control group. (A) ROS generation was measured using oxidation-sensitive fluorescent probe (DHE) by flow cytometry. Means and S.D.s of three repeats were shown. (B) HO8910 and OVCAR3 cells were exposed to 5, 15, and 30μM curcumin for 18h. 0.1% DMSO was used as control. IF images showing the γH2AX; Hoechst was used for the nuclear staining. Scale bar, 53.5 μm and 96.9μm.

induce G2/M arrest, and their anti-tumor effects are at least partially mediated by G2/M arrest associated oxidative stress [20,22].

Mitochondria are the major source of ROS production. Mitochondrial oxidative phosphorylation has been found to be altered in ovarian cancer and mitochondria have been suggested as a therapeutic target [20,31]. Cancer cells exhibit a high degree of metabolic heterogeneity [30] and resistance to apoptosis [15] and it has been shown that cancer cells can switch to OXPHOS metabolism in the process of acquiring therapeutic resistance [32]. A by-product of oxidative phosphorylation (OXPHOS) is the formation of $O_2^{\bullet-}$ and $H_2O_2$, especially $O_2^{\bullet-}$,

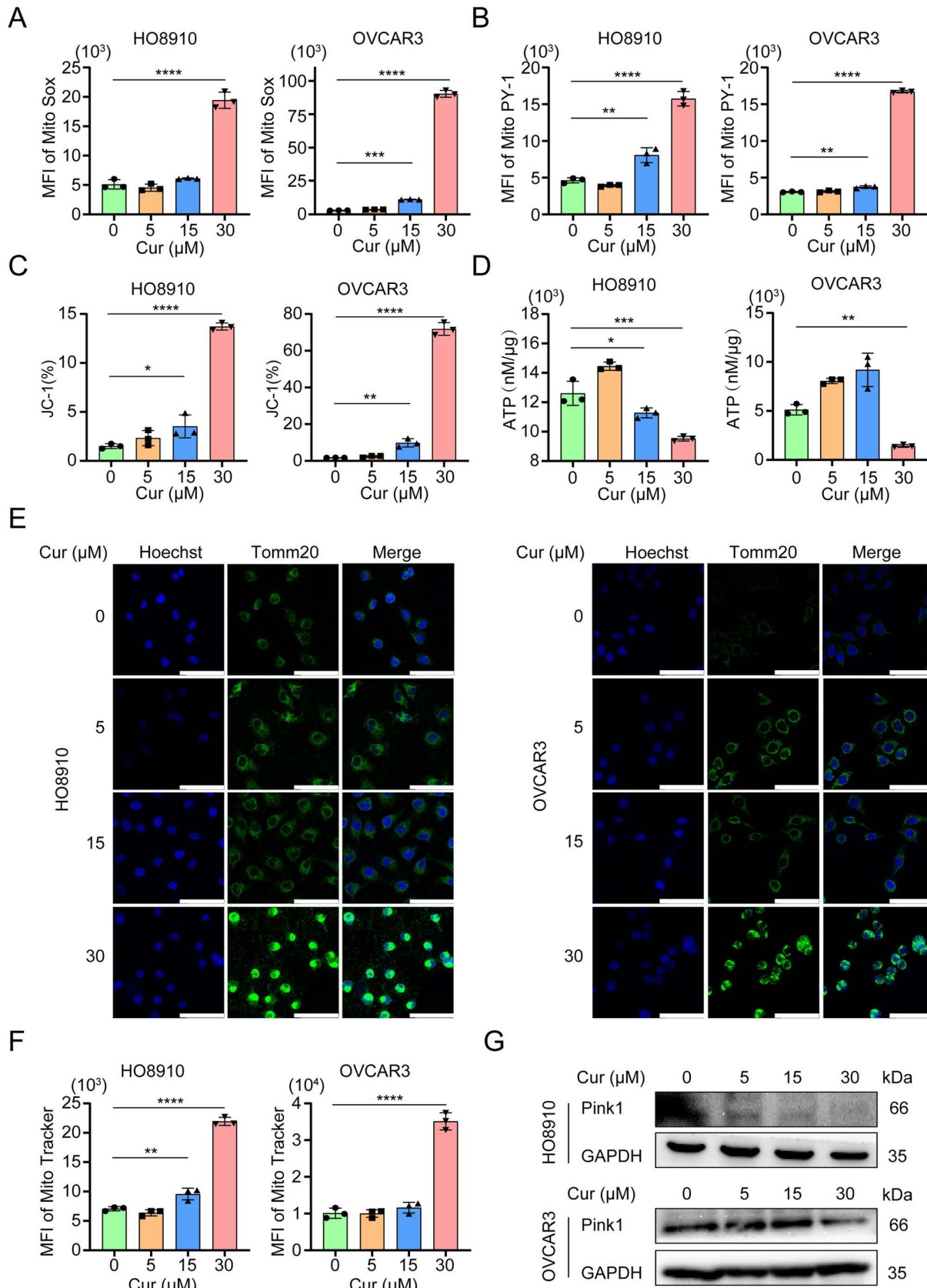

**Fig 4. Curcumin induces mitochondrial stress in ovarian cancer cells.** HO8910 and OVCAR3 cells were exposed to 5, 15, and 30μM curcumin for 48h. 0.1% DMSO was used as control. Means and S.D.s of three repeats were shown. $* p < 0.05$, $** p < 0.01$, $*** p < 0.001$,

****$p < 0.0001$. (A) O2·- generation was measured using fluorescent probe Mito Sox by flow cytometry. (B) H2O2 generation was measured using fluorescent probe Mito PY-1 by flow cytometry. (C) Mitochondrial membrane potential was measured using JC-1 probe by flow cytometry. (D) ATP generation was measured using ATP assay kit. (E) IF images showing Tomm20; Hoechst was used for the nuclear staining. Scale bar, 61.5 μm. (F) Mitochondrial amount was measured using fluorescent probe Mito Tracker Red by flow cytometry. (G) Whole cell lysates were analyzed by immunoblotting with antibodies specific for PINK1. GAPDH was used as a loading control.

whereas $H_2O_2$ is produced as a product of superoxide dismutase (SOD2) [33]. This is particularly relevant in ovarian cancer [34], where only 10-30% of patients achieve a complete response to initial therapy [35]. In this study, we demonstrated that in curcumin-treated ovarian cancer cells, total ROS, $O_2^{\cdot-}$, $H_2O_2$ and •OH were significantly increased, mitochondrial membrane potential was significantly decreased, ATP production was greatly reduced, and the mitochondrial mass was drastically increased but their function was impaired. The significant increase in Tomm20 indicates the accumulation of mitochondrial precursors in the cytoplasm [21]. Disruption of DNA repair protein RAD51 in ovarian cancer cells also leads to G2/M arrest, excessive accumulation of dysfunctional mitochondria and mitochondrial superoxide [20]. Spindle-targeting cancer therapeutics similarly induce G2/M arrest coupled mitochondrial stress that mediates the anti-tumor effect [22]. Double-strand breaks (DSBs) are one of the most deleterious types of DNA damage and can cause death if left unrepaired. They are usually repaired by non-homologous end-joining and homologous recombination. Homologous recombination is the primary repair pathways for DSBs occurring during the G2 phase of cells.

These findings suggest that mitochondria may serve as a hub that senses and transmits the stress signals in cells under adverse conditions. Mitochondrial autophagy is divided into two pathways, of which PINK1 is the major protein of the non-ubiquitination pathway. Our study showed a dramatic increase in the number of mitochondria and a significant down-regulation of PINK1 expression in curcumin-treated ovarian cancer cells, suggesting that the accumulation of dysfunctional mitochondria might be due to a decrease in mitochondrial autophagy.

Curcumin has been reported to possess antioxidant effect in many other pathophysiological settings. For example, it inhibits oxidative stress in hepatocytes through activation of PPAR-α [36]. Curcumin ameliorates aortic endothelial injury by reducing ROS [37]. It also ameliorates endothelial damage in rat retina by inhibiting ROS production [38]. Curcumin was shown to reduce ROS levels, mainly by restoring NO levels, inhibiting endothelial inflammation and oxidative damage, and reducing leukocyte adhesion and endothelial cell apoptosis [39]. Consistently, ROS level and DNA damage were not elevated in ovarian epithelial cells in this study. Thus, the oxidative stress-inducing effect of curcumin is probably specific to cancer cells.

Previous studies have established that curcumin is not toxic to humans when taken at a daily intake of 8 g. Additionally, a phase 2 clinical trial was carried out to evaluate curcumin's potential role in the prevention of colorectal tumors [40]. Evidence from clinical practice indicates that the effect of co-treatment with antioxidants during chemotherapy is unpredictable and may result in adverse outcomes [41–47]. The differences in body weight of nude mice between treatment group and control group in this study was not statistically significant, indicating that curcumin is safe along with its anti-tumor effect. The findings reveal a novel anti-tumor mechanism of curcumin, a traditional natural compound, and provides a new idea for the fight against cancer.

To date, more than 230 clinical trials have been investigated to understand the pharmacological aspects of curcumin in the human system [43]. Although curcumin supplements are one of the best-selling botanicals with encouraging preclinical results, its bioactivity remains a concern [48], which is related to its low solubility in water, low stability under physiological

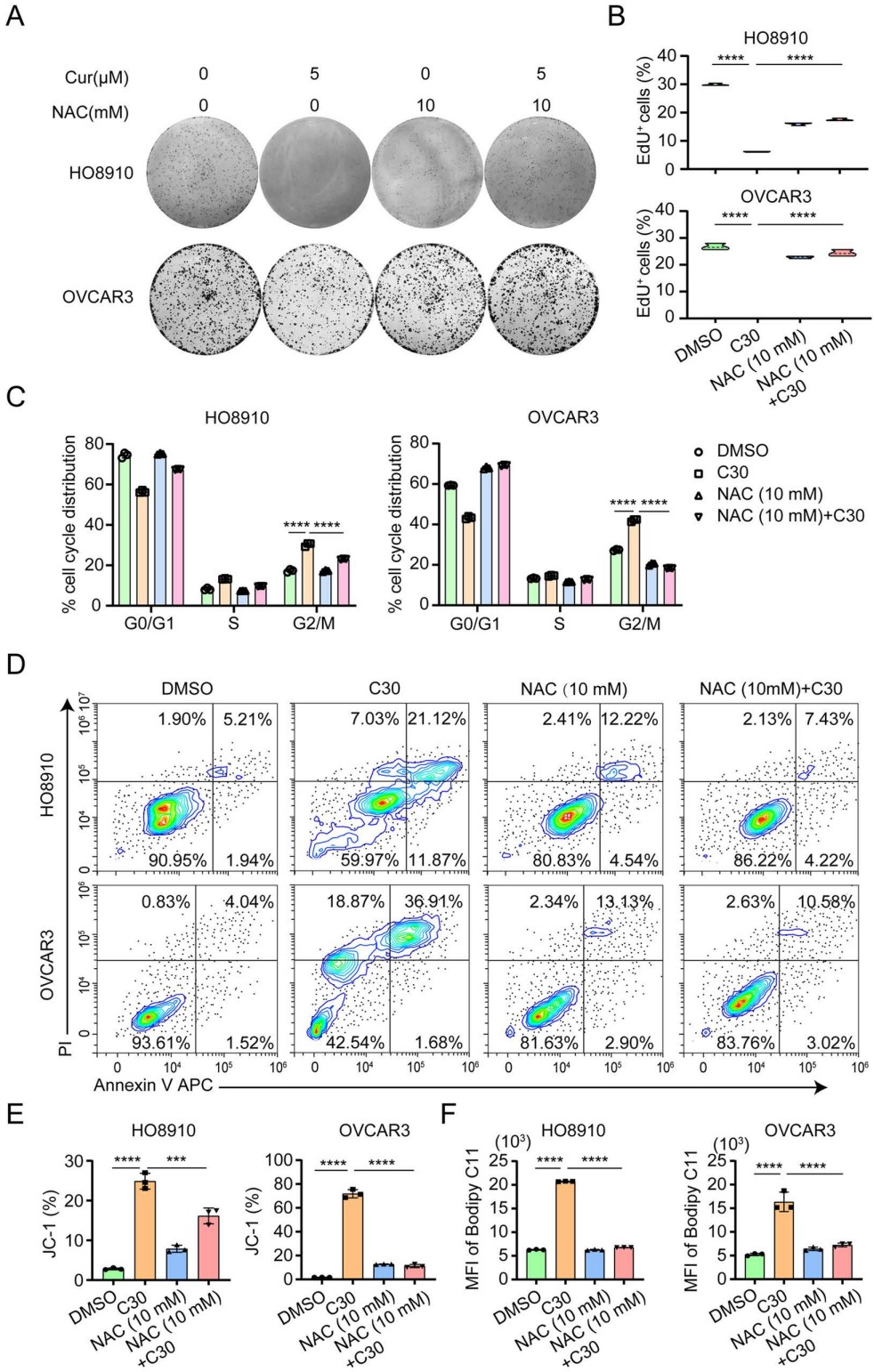

**Fig 5. NAC rescues the cytotoxic effect of curcumin on ovarian cancer cells.** HO8910 and OVCAR3 cells were treated with curcumin, NAC alone or in combination(curcumin was add to into medium 4h after NAC). (A) Colony formation assay were observed for 8-10 days. (B) EdU proliferation assay was performed by using EdU assay kit. (C) The distribution of cell cycle was assessed by flow cytometry. (D) Apoptosis was measured by flow cytometry with annexin V and PI staining. (E) Mitochondrial membrane potential alteration was measured using JC-1 probe by flow cytometry. (F) Bodipy C11 probe was used for measuring lipid peroxidation by flow cytometry.

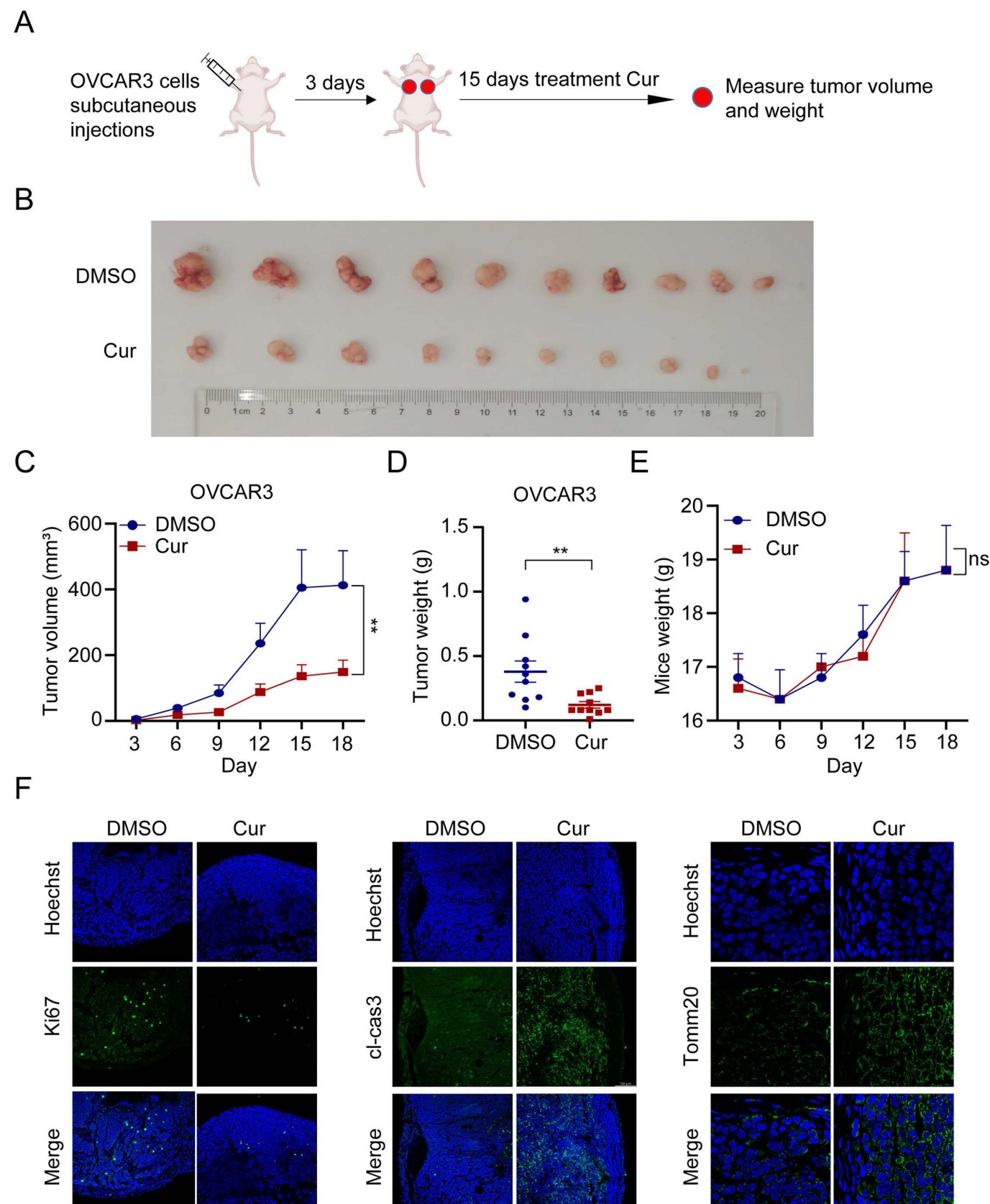

**Fig 6. Curcumin inhibits tumour growth in vivo.** (A) Scheme for the treatment paradigm.10 4-6-week nude mice were randomly divided into two groups, each group had 5 mice. (B) Images of OVCAR3 tumors for each treatment group. (C) Growth curves of tumors from transplanted OVCAR3 cells in nude mice

for each treatment group. (D) Average tumor weight on day 21 for each treatment group. (E) Weight curves of mice from transplanted OVCAR3 cells in nude mice for each treatment group. (F) Representative IF images showing the Ki67, cleaved caspase-3, Scale bar, 100 μm.Tomm20, Scale bar, 50 μm. Hoechst was used for the nuclear staining.

pH conditions, rapid metabolism and poor blood-brain barrier permeability [48]. Thus, the translation of the in vitro findings with curcumin needs to be further explored. Due to the problems associated with the bioavailability of curcumin and off-target effects such as nausea and diarrhea, the application of nanotechnology that improves the bioavailability is expected to offset the problems associated with the bioavailability of curcumin.

## Conclusion

Curcumin exerts anti-neoplastic effects by inducing oxidative stress, DNA damage and mitochondrial dysfunction in ovarian cancer cells, offered novel approach to treating ovarian cancer.

## Supporting information

**S1 Fig. Effect of curcumin in ovarian epithelium IOSE80 cells.** (A) IOSE80 cells were exposed to curcumin (2-45 μM) or vehicle control (0.1% DMSO) for 24 h, 48 h and 72 h. Cell viability was measured by CCK-8 assay. The experiments were performed in quadruplicate. (B) IOSE80 cells were exposed to 5, 15 and 30 μM curcumin for 48 h. 0.1% DMSO was used as control. $*p < 0.05$, $**p < 0.01$, $***p < 0.001$, $****p < 0.0001$, when compared with control group. ROS generation was measured using oxidation-sensitive fluorescent probe (DHE) by flow cytometry. Means and S.D.s of three repeats were shown. (C) IOSE80 cells were exposed to 5, 15 and 30 μM curcumin for 48 h. 0.1% DMSO was used as control. Cells were processed by flow cytometry using Annexin V and PI staining. Results shown are representative of three independent experiments. (D) IOSE80 cells were exposed to 5, 15 and 30μM curcumin for 12 h. 0.1% DMSO was used as control. IF images showing the γ-H2AX; Hoechst was used for the nuclear staining. Scale bar, 100 μm.
(TIF)

**S2 Fig. Low dose curcumin fails to inhibit proliferation of ovarian cancer cells.** HO8910 cells were exposed to curcumin (0.25-4 μM) or vehicle control (0.1% DMSO) for 24 h, 48 h and 72h. Cell viability was measured by CCK-8 assay. The experiments were performed in quadruplicate.
(TIF)

**S1 File. All raw blot and gel images.**
(PDF)

## Acknowledgements

We thank Drs. Yufang Shi and Yongjing Chen for logistic support.

## Author contributions

**Data curation:** Qi Bao, Zihan Wang, Tingting Yang, Ying Chen, Qicheng Deng.

**Formal analysis:** Qi Bao, Xiao Su, Ying Chen.

**Funding acquisition:** Changshun Shao, Weipei Zhu.

**Investigation:** Qingyang Liu, Weipei Zhu.

**Methodology:** Qi Bao, Zihan Wang, Tingting Yang, Xiao Su, Lifen Liu, Qicheng Deng, Weipei Zhu.

**Project administration:** Changshun Shao.

**Resources:** Qi Bao.

**Software:** Xiao Su, Lifen Liu.

**Supervision:** Changshun Shao, Weipei Zhu.

**Writing – original draft:** Qi Bao, Zihan Wang.

**Writing – review & editing:** Changshun Shao, Weipei Zhu.

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
