## [Decision Letter · Decision Letter 0]

30 Oct 2024

PONE-D-24-40005

Curcumin induces mitochondrial dysfunction-associated oxidative DNA damage in ovarian cancer cells

PLOS ONE

Dear Dr. Zhu,

Thank you for submitting your manuscript to PLOS ONE. After careful consideration, we feel that it has merit but does not fully meet PLOS ONE’s publication criteria as it currently stands. Therefore, we invite you to submit a revised version of the manuscript that addresses the points raised during the review process.

My comments are included as the second reviewer.

We look forward to receiving your revised manuscript.

Kind regards,

Sanaz Alaeejahromi

Academic Editor

PLOS ONE

“Weipei Zhu. Second Affiliated Hospital of Soochow University（XKTJ-XK202006）.

Changshun Shao. National Natural Science Foundation of China (32150710523”

6. Please include a separate caption for each figure in your manuscript.

7. PLOS ONE now requires that authors provide the original uncropped and unadjusted images underlying all blot or gel results reported in a submission’s figures or Supporting Information files. This policy and the journal’s other requirements for blot/gel reporting and figure preparation are described in detail at https://journals.plos.org/plosone/s/figures#loc-blot-and-gel-reporting-requirements and https://journals.plos.org/plosone/s/figures#loc-preparing-figures-from-image-files. When you submit your revised manuscript, please ensure that your figures adhere fully to these guidelines and provide the original underlying images for all blot or gel data reported in your submission. See the following link for instructions on providing the original image data: https://journals.plos.org/plosone/s/figures#loc-original-images-for-blots-and-gels.  

Reviewers' comments:

Reviewer's Responses to Questions

**Comments to the Author**

1. Is the manuscript technically sound, and do the data support the conclusions?

Reviewer #1: Partly

Reviewer #2: Yes

2. Has the statistical analysis been performed appropriately and rigorously?

Reviewer #1: No

Reviewer #2: I Don't Know

3. Have the authors made all data underlying the findings in their manuscript fully available?

Reviewer #1: Yes

Reviewer #2: No

4. Is the manuscript presented in an intelligible fashion and written in standard English?

Reviewer #1: No

Reviewer #2: Yes

5. Review Comments to the Author

Reviewer #1: Issue: The study investigates curcumin's effect on ovarian cancer cells, which is a well-explored area. Many studies have already demonstrated curcumin’s anticancer properties, particularly its role in inducing oxidative stress, apoptosis, and mitochondrial dysfunction. The manuscript doesn't clearly state what sets this research apart from existing studies or what unique contributions it makes to the field.

Abstract:

The abstract is concise and covers key aspects of the study. However, it could be enhanced by explicitly stating the novelty or unique findings of the research. For instance, mentioning more specific implications for clinical treatment could provide clarity.

Consider adding brief details about the doses used and results from in vivo models for a more complete overview.

Introduction: While the rationale for using curcumin is well presented, it might benefit from more information about the gap in current literature that the study aims to address. The introduction could be improved by stating the specific hypotheses or questions being tested more explicitly

The methods section does not explicitly mention how the different concentrations of curcumin (2 to 45 μM) were selected. Typically, drug concentrations like curcumin are chosen based on previous studies and determined by the IC50 or effective dose ranges in laboratory experiments. It is likely that the researchers selected these concentrations based on prior research findings or pilot studies to assess dose-dependent effects. To provide a more precise explanation in the manuscript, it would be helpful for the authors to clarify how these concentrations were chosen, for example, based on previous studies or preliminary experiments to determine the IC50. This information could contribute to a better understanding of the results.

the methods section could benefit from more specific details on the statistical methods used to ensure reproducibility. Including the number of technical and biological replicates for each experiment (e.g., for flow cytometry) would improve transparency.

When using ANOVA to compare more than two groups, post-hoc tests (e.g., Tukey’s test or Bonferroni correction) are necessary to identify specific group differences. The manuscript does not mention whether such post-hoc tests were performed. Omitting this step can result in incomplete analysis when significant differences are detected by ANOVA.

While the results are comprehensive, there is a lack of clear narrative flow between the experiments. For instance, the connection between the mitochondrial dysfunction in vitro and tumor suppression in vivo isn’t fully explained. Some results, like the decreased ROS levels in non-cancerous cells, are intriguing but aren’t discussed adequately.

Comparison: Well-organized manuscripts have a clear logical flow from one result to the next, and each finding is discussed in sufficient depth. A stronger explanation of the significance of each result and its connection to the overall hypothesis would enhance the manuscript’s coherence.

The manuscript does not adequately address the study’s limitations, such as the relevance of in vitro findings to clinical settings or the potential off-target effects of curcumin. A lack of critical reflection can suggest a biased interpretation of the results.

Reviewer #2: The research examines the impact of curcumin on ovarian cancer cells, a topic that has been extensively studied. Numerous investigations have previously validated curcumin's anticancer effects, especially regarding its ability to promote oxidative stress, induce apoptosis, and cause mitochondrial dysfunction. However, the manuscript fails to clearly articulate how this study distinguishes itself from previous research or what novel insights it offers to the field.

The abstract effectively summarizes the primary components of the study; however, it would benefit from a clearer emphasis on the unique contributions of the research.

The introduction lays a solid foundation for the rationale behind curcumin's use.

Methods: The methods section lacks clarity regarding the rationale for selecting the curcumin concentrations (ranging from 2 to 45 μM).

It would be beneficial for the authors to explain whether these concentrations were derived from previous studies or preliminary experiments.

Furthermore, the section could improve by detailing the statistical analyses employed, including the number of biological and technical replicates for each experiment, such as those conducted via flow cytometry.

Results: Although the results section is thorough, it lacks a cohesive narrative that connects the various experiments.

Some findings, such as the reduction of ROS levels in non-cancerous cells, are compelling but warrant further discussion to contextualize their significance.

---

## [Author Response · Author response to Decision Letter 1]

25 Nov 2024

Responses to reviews (original comments by reviews are in blue color)

Reviewer # 1:

1.Comment: Issue: The study investigates curcumin's effect on ovarian cancer cells, which is a well-explored area. Many studies have already demonstrated curcumin’s anticancer properties, particularly its role in inducing oxidative stress, apoptosis, and mitochondrial dysfunction. The manuscript doesn't clearly state what sets this research apart from existing studies or what unique contributions it makes to the field.

1.Reply: Thank you so much for your comments on our manuscript. The anticancer properties of curcumin, especially its role in inducing oxidative stress, apoptosis and mitochondrial dysfunction, have indeed been demonstrated in many studies. However, this effect of curcumin on ovarian cancer cells has not been well characterized. Previous studies tend to concentrate on other aspects of the cellular response to curcumin, as shown below:

Ravindran F, Mhatre A, Koroth J, Narayan S, Choudhary B. Curcumin modulates cell type-specific miRNA networks to induce cytotoxicity in ovarian cancer cells. Life Sci. 2023 Dec 1;334:122224. doi: 10.1016/j.lfs.2023.122224. Epub 2023 Oct 30. PMID: 38084671.

Liu Y, Shen Z, Zhu T, Lu W, Fu Y. Curcumin enhances the anti-cancer efficacy of paclitaxel in ovarian cancer by regulating the miR-9-5p/BRCA1 axis. Front Pharmacol. 2023 Jan 10;13:1014933. doi: 10.3389/fphar.2022.1014933. PMID: 36703740; PMCID: PMC9871306.

Zheng N, Liu S, Zeng H, Zhao H, Jin L. Molecular Mechanism of Curcumin Derivative on YAP Pathway against Ovarian Cancer. J Clin Med. 2022 Dec 5;11(23):7220. doi: 10.3390/jcm11237220. PMID: 36498794; PMCID: PMC9740724. Our study shows that curcumin treatment of ovarian cancer cells induces mitochondrial dysfunction-associated elevated ROS and DNA damage, ultimately leading to their apoptotic fate.

2. Comment: Abstract: The abstract is concise and covers key aspects of the study. However, it could be enhanced by explicitly stating the novelty or unique findings of the research. For instance, mentioning more specific implications for clinical treatment could provide clarity.

Consider adding brief details about the doses used and results from in vivo models for a more complete overview.

2.Reply: We cited literature “ Tate, J.G., et al., COSMIC: the Catalogue Of Somatic Mutations In Cancer. Nucleic Acids Res, 2019. 47(D1): p. D941-D947.” as reference [2]. We added “This effect of curcumin is independent of the BRCA mutation status” in Abstract. We also added “ According to the Catalogue Of Somatic Mutations In Cancer (COSMIC), ovarian cancer cell lines such as OVCAR3, A2780, SKOV3, HEY, etc. are free of BRCA mutation, and only HO8910 carries BRCA1 5382C mutation. Therefore, the HO8910 cell line and the OVCAR3 cell line were selected for this study” on page 3. Consider adding brief details about the doses used and results from in vivo models for a more complete overview. We added “ 200 mg/kg intraperitoneal infusion daily ” and “ oxidative damage, and mitochondrial accumulation ” on page 2. We added: and can be potentially used in combination with other DNA repair-interfering therapeutics, such as PARP inhibitor, in the treatment of ovarian cancer at the end of the abstract.

3.Comment: Introduction: While the rationale for using curcumin is well presented, it might benefit from more information about the gap in current literature that the study aims to address. The introduction could be improved by stating the specific hypotheses or questions being tested more explicitly.

3.Reply: We added “While an anticancer effect of curcumin was also reported on ovarian cancer, whether or not curcumin induces mitochondrial dysfunction has not been clarified [10, 13, 14], and”. they were supplemented by citations to the relevant literature on page 4.

4.Comment: The methods section does not explicitly mention how the different concentrations of curcumin (2 to 45 μM) were selected. Typically, drug concentrations like curcumin are chosen based on previous studies and determined by the IC50 or effective dose ranges in laboratory experiments. It is likely that the researchers selected these concentrations based on prior research findings or pilot studies to assess dose-dependent effects. To provide a more precise explanation in the manuscript, it would be helpful for the authors to clarify how these concentrations were chosen, for example, based on previous studies or preliminary experiments to determine the IC50. This information could contribute to a better understanding of the results.

4.Reply: We selected the different concentrations of curcumin (2 to 45 μM) were based on previous studies or preliminary experiments. At first, we used a low concentration of curcumin and found that curcumin had no effect on the viability of ovarian cancer cells below a concentration of 5 μM, so the curcumin concentration was increased to more than 5 μM for the IC 50 assay, as shown in figure below ( and we submitted this result as Figure S2).

5.Comment: the methods section could benefit from more specific details on the statistical methods used to ensure reproducibility. Including the number of technical and biological replicates for each experiment (e.g., for flow cytometry) would improve transparency.

5.Reply: We added “ The experiment was repeated 3 times and each contained at leasthree biological replicates.” on page 6. The other experiments e.g. EdU proliferation assay, we discribed flow cytometry as “Ratio of EdU positive cells is shown as the mean ± S.D. from three independent experiments.” And we added “ The experiment was repeated 3 times and each contained at leasthree biological replicates.” on page 6, 7, 8, 9, 10 “Material and method ”.

6.Comment: When using ANOVA to compare more than two groups, post-hoc tests (e.g., Tukey’s test or Bonferroni correction) are necessary to identify specific group differences. The manuscript does not mention whether such post-hoc tests were performed. Omitting this step can result in incomplete analysis when significant differences are detected by ANOVA.

6.Reply:Tukey’s test was used to determine the statistical significance between experimental groups. So we added “and Tukey’s test were was” on page 10.

7.Comment: While the results are comprehensive, there is a lack of clear narrative flow between the experiments. For instance, the connection between the mitochondrial dysfunction in vitro and tumor suppression in vivo isn’t fully explained. Some results, like the decreased ROS levels in non-cancerous cells, are intriguing but aren’t discussed adequately.

Comparison: Well-organized manuscripts have a clear logical flow from one result to the next, and each finding is discussed in sufficient depth. A stronger explanation of the significance of each result and its connection to the overall hypothesis would enhance the manuscript’s coherence.

7.Reply: Thank you for your wise suggestions! To well-organized one result to the next, we added “During cell culture, we found that curcumin-treated ovarian cancer cells tended to float and the number of floating cells increased with increasing curcumin concentration.” on page 12. “The aforementioned experiments revealed that curcumin inhibits the proliferation and induces apoptosis of ovarian cancer cells by up-regulating oxidative stress, and to substantiate this notion, we attempted a rescue experiment with the antioxidant NAC” on page 14, 15. “We further explored whether the tumour-suppressive effects and mechanisms of curcumin in in vivo experiments were consistent with those in vitro.” on page 15.

Some literature revealed that curcumin can inhibit ROS levels and oxidative stress in non-cancerous cells, e.g. “Curcumin has been reported to possess antioxidant effect in many other pathophysiological settings. For example, it inhibits oxidative stress in hepatocytes through activation of PPAR-α[31]. Curcumin ameliorates aortic endothelial injury by reducing ROS [32]. It also ameliorates endothelial damage in rat retina by inhibiting ROS production[33]. Curcumin was shown to reduce ROS levels, mainly by restoring NO levels, inhibiting endothelial inflammation and oxidative damage, and reducing leukocyte adhesion and endothelial cell apoptosis[34]. Consistently, ROS level and DNA damage were not elevated in ovarian epithelial cells. Thus, the oxidative stress-inducing effect of curcumin is probably specific to cancer cells.” in our study is consistent with the results of the previous study on page 18, 19.

8.Comment: The manuscript does not adequately address the study’s limitations, such as the relevance of in vitro findings to clinical settings or the potential off-target effects of curcumin. A lack of critical reflection can suggest a biased interpretation of the results.

8.Reply: Considering the limitations of this study and the off-target effects of curcumin. We added “To date, more than 230 clinical trials have been investigated to understand the pharmacological aspects of curcumin in the human system[43]. Although curcumin supplements are one of the best-selling botanicals with encouraging preclinical results, questions remain about bioactivity in human body [44].Curcumin has low solubility in water, is unstable under physiological pH conditions and is rapidly metabolised, which leads to its low oral bioavailability and poor blood-brain barrier permeability[45]. Due to the problems associated with the bioavailability of curcumin and off-target effects such as nausea and diarrhea, the application of nanotechnology to improve bioavailability and mitigate side effects could be further explored and developed.” And they were supplemented by citations to the relevant literature on page 19, 20.

Reviewer # 2:

1.Comment: The research examines the impact of curcumin on ovarian cancer cells, a topic that has been extensively studied. Numerous investigations have previously validated curcumin's anticancer effects, especially regarding its ability to promote oxidative stress, induce apoptosis, and cause mitochondrial dysfunction. However, the manuscript fails to clearly articulate how this study distinguishes itself from previous research or what novel insights it offers to the field.

1.Reply: 1.Reply: Thank you so much for your comments on our manuscript. The anticancer properties of curcumin, especially its role in inducing oxidative stress, apoptosis and mitochondrial dysfunction, have indeed been demonstrated in many studies. However, this effect of curcumin on ovarian cancer cells has not been well characterized. Previous studies tend to concentrate on other aspects of the cellular response to curcumin, as shown below:

Ravindran F, Mhatre A, Koroth J, Narayan S, Choudhary B. Curcumin modulates cell type-specific miRNA networks to induce cytotoxicity in ovarian cancer cells. Life Sci. 2023 Dec 1;334:122224. doi: 10.1016/j.lfs.2023.122224. Epub 2023 Oct 30. PMID: 38084671.

Liu Y, Shen Z, Zhu T, Lu W, Fu Y. Curcumin enhances the anti-cancer efficacy of paclitaxel in ovarian cancer by regulating the miR-9-5p/BRCA1 axis. Front Pharmacol. 2023 Jan 10;13:1014933. doi: 10.3389/fphar.2022.1014933. PMID: 36703740; PMCID: PMC9871306.

Zheng N, Liu S, Zeng H, Zhao H, Jin L. Molecular Mechanism of Curcumin Derivative on YAP Pathway against Ovarian Cancer. J Clin Med. 2022 Dec 5;11(23):7220. doi: 10.3390/jcm11237220. PMID: 36498794; PMCID: PMC9740724. Our study proposes that curcumin treatment of ovarian cancer cells induces mitochondrial dysfunction-associated elevated ROS and DNA damage, ultimately leading to their apoptotic fate.

2.Comment: The abstract effectively summarizes the primary components of the study; however, it would benefit from a clearer emphasis on the unique contributions of the research.

2.Reply: We cited literature “ Tate, J.G., et al., COSMIC: the Catalogue Of Somatic Mutations In Cancer. Nucleic Acids Res, 2019. 47(D1): p. D941-D947.” as reference [2]. We added This effect of curcumin is independent of the BRCA mutation status in Abstract. We also added “ According to the Catalogue Of Somatic Mutations In Cancer (COSMIC), ovarian cancer cell lines such as OVCAR3, A2780, SKOV3, HEY, etc. are free of BRCA mutation, and only HO8910 carries BRCA1 5382C mutation. Therefore, the HO8910 cell line and the OVCAR3 cell line were selected for this study” on page 3. Consider adding brief details about the doses used and results from in vivo models for a more complete overview. We added “ 200 mg/kg intraperitoneal infusion daily ” and “ oxidative damage, and mitochondrial accumulation ” on page 2. We added: and can be potentially used in combination with other DNA repair-interfering therapeutics, such as PARP inhibitor, in the treatment of ovarian cancer at the end of the abstract.

3.Comment: The introduction lays a solid foundation for the rationale behind curcumin's use. Methods: The methods section lacks clarity regarding the rationale for selecting the curcumin concentrations (ranging from 2 to 45 μM).

It would be beneficial for the authors to explain whether these concentrations were derived from previous studies or preliminary experiments.

3.Reply: We selected the different concentrations of curcumin (2 to 45 μM) were based on previous studies or preliminary experiments. At first, we used a low concentration of curcumin and found that curcumin had no effect on the viability of ovarian cancer cells below a concentration of 5 μM, so the curcumin concentration was increased to more than 5 μM for the IC 50 assay, as shown in figure below ( and we presented this result as Figure S2).

3.Comment: Furthermore, the section could improve by detailing the statistical analyses employed, including the number of biological and technical replicates for each experiment, such as those conducted via flow cytometry.

4.Reply: We added “Three technical and biological replicates per experiment at least. ” and “and Tukey’s test were” on Page 10.

5.Comment: Results: Although the results section is thorough, it lacks a cohesive narrative that connects the various experiments.

5.Reply: Thank you for your wise suggestions! To well-organized one result to the next, we added “During cell culture, we found that curcumin-treated ovarian cancer cells tended to float and the number of floating cells increased with increasing curcumin concentration.” on page 12. “The aforementioned experiments revealed that curcumin inhibits the proliferation and induces apoptosis of ovarian cancer cells by up-regulating oxidative stress, and to substantiate this notion, we attempted a rescue experiment with the antioxidant NAC” on page 14, 15. “We further explored whether the tumour-suppressive effects and mechanisms of curcumin in in vivo experiments were consistent with those in vitro.” on page 15.

6.Comment: Some findings, such as the reduction of ROS levels in non-cancerous cells, are compelling but warrant further discussion to contextualize their significance.

6.Reply: Some literature revealed that curcumin can inhibit ROS levels and oxidative stress in non-cancerous cells, e.g. “Curcumin has been reported to possess antioxidant effect in many other pathophysiological settings. For example, it inhibits oxidative stress in hepatocytes through activation of PPAR-α[31]. Curcumin ameliorates aortic endothelial injury by reducing ROS [32]. It also ameliorates endothelial damage in rat retina by inhibiting ROS production[33]. Curcumin was shown to reduce ROS levels, mainly by restoring NO levels, inhibiting endothelial inflammation and oxidative damage, and reducing leukocyte adhesion and endothelial cell apoptosis[34]. Consistently, ROS level and DNA damage were not elevated in ovarian epithelial cells. Thus, the oxidative stress-inducing effect of curcumin is probably specific to cancer cells.” in our study is consistent with the results of the previous study on page 18, 19.

We are grateful for the invaluable feedback provided by the reviewers, and we look forward to the opportunity to further improve our manuscript.

With kind regards,

Weipei Zhu, PhD

---

## [Decision Letter · Decision Letter 1]

20 Jan 2025

PONE-D-24-40005R1Curcumin induces mitochondrial dysfunction-associated oxidative DNA damage in ovarian cancer cellsPLOS ONE

Dear Dr. Zhu,

Thank you for submitting your manuscript to PLOS ONE. After careful consideration, we feel that it has merit but does not fully meet PLOS ONE’s publication criteria as it currently stands. Therefore, we invite you to submit a revised version of the manuscript that addresses the points raised during the review process.

We look forward to receiving your revised manuscript.

Kind regards,

Sana Alaeejahromi

Academic Editor

PLOS ONE

Journal Requirements:

Reviewers' comments:

Reviewer's Responses to Questions

**Comments to the Author**

1. If the authors have adequately addressed your comments raised in a previous round of review and you feel that this manuscript is now acceptable for publication, you may indicate that here to bypass the “Comments to the Author” section, enter your conflict of interest statement in the “Confidential to Editor” section, and submit your "Accept" recommendation.

Reviewer #1: All comments have been addressed

Reviewer #3: All comments have been addressed

2. Is the manuscript technically sound, and do the data support the conclusions?

Reviewer #1: Yes

Reviewer #3: Yes

3. Has the statistical analysis been performed appropriately and rigorously? 

Reviewer #1: Yes

Reviewer #3: I Don't Know

4. Have the authors made all data underlying the findings in their manuscript fully available?

Reviewer #1: Yes

Reviewer #3: Yes

5. Is the manuscript presented in an intelligible fashion and written in standard English?

Reviewer #1: Yes

Reviewer #3: Yes

6. Review Comments to the Author

Reviewer #1: (No Response)

Reviewer #3: The authors investigated that curcumin can exert its anti-tumor effect via inducing mitochondrial dysfunction-associated oxidative DNA damage and can be potentially used in combination with other DNA repair-interfering therapeutics, such as PARP inhibitor, in the treatment of ovarian cancer.

The authors have performed a comprehensive analysis and efficiently conveyed their findings. Nevertheless, there are certain inquiries and recommendations that, if taken into consideration, could improve the overall caliber of the manuscript. Below are some comments about this study:

Abstract

The abstract summarizes the study’s structure, methods, and main results.

Introduction

1- Comment: First add some other examples of the beneficial effects of curcumin in reproductive system. Then follow with specifically about its role in reducing oxidative stress in female reproductive tissues under similr toxicological conditions. Use the below studies to complete these parts:

Protective effects of curcumin on chromatin quality, sperm parameters, and apoptosis following testicular torsion-detorsion in mice

Curcumin mitigates acrylamide‐induced ovarian antioxidant disruption and apoptosis in female Balb/c mice: A comprehensive study on gene and protein expressions

Curcumin and Its Nanoformulations: Exploring Therapeutic Potential in Female Reproductive Health

2- Comment: While the rationale for using curcumin is well presented, it might benefit from more information about the gap in current literature that the study aims to address. The introduction could be improved by stating the specific hypotheses or questions being tested more explicitly.

3- Comment: Please delete the last paragraph in the introduction “In this study we observed elevated oxidative stress, increased DNA damage, elevated mitochondrial reactive oxygen species (ROS) levels, decreased ATP production, increased mitochondrial number, and decreased mitochondrial autophagy in curcumin-treated ovarian cancer cells, indicating that curcumin might exert its anti-tumor effect via inducing mitochondrial stress”, this is the results of your study and it is better to add in the result or discussion.

Comment: The introduction is clear and well-organized but. It is suggested to write the objective of the study as below: whether or not curcumin induces mitochondrial dysfunction has not been defined [10, 13, 14], and the exact processes underpinning curcumin-induced oxidative stress and DNA damage are still to be thoroughly characterized. Our research may provide new insights into how curcumin may exert its anti-tumor effect by inducing mitochondrial stress.

Materials and Methods

It is written in detail and completely.

“Curcumin treatment of cultured cells Curcumin (S1848, Selleck Chemicals) was dissolved in DMSO at a stock con centration of 50 mM and applied to cells in culture at different final concentrations.”

Would you please clarify which cell concentration you used?

Results

It is recommended to delete any numbers and percentages in the text of the results that is shown in figures.

Discussion

It is written comprehensively. Start the first paragraph of the discussion with the main findings of the study.

I appreciate the opportunity provided by the esteemed editor to evaluate this manuscript. I recommend its minor revision. I read it carefully.

This study is a valuable and well-conducted investigation, providing insightful contributions to the understanding of toxicology and reproductive health. The research is methodologically sound, with strong experimental design and clear, impactful conclusions. The authors effectively demonstrate the potential therapeutic role of curcumin, making this study both relevant and promising for future clinical applications. Overall, the work is of high quality and advances current knowledge in the field, offering novel approach to treating ovarian cancer.

comment: The manuscript does not adequately address the study’s limitations, such as the relevance of in vitro findings to clinical settings or the potential off-target effects of curcumin. A lack of critical reflection can suggest a biased interpretation of the results.

Conclusion:

There is no clear conclusion section. Please add it

Best Regards

7. PLOS authors have the option to publish the peer review history of their article (what does this mean? ). If published, this will include your full peer review and any attached files.

**Do you want your identity to be public for this peer review?** For information about this choice, including consent withdrawal, please see our Privacy Policy .

Reviewer #1: No

Reviewer #3: No

---

## [Author Response · Author response to Decision Letter 2]

5 Feb 2025

Responses to reviews (original comments by reviews are in blue color)

Reviewer # 3:

The authors have performed a comprehensive analysis and efficiently conveyed their findings. Nevertheless, there are certain inquiries and recommendations that, if taken into consideration, could improve the overall caliber of the manuscript. Below are some comments about this study:

Abstract

The abstract summarizes the study’s structure, methods, and main results.

Introduction

1- Comment: First add some other examples of the beneficial effects of curcumin in reproductive system. Then follow with specifically about its role in reducing oxidative stress in female reproductive tissues under similr toxicological conditions. Use the below studies to complete these parts:

Protective effects of curcumin on chromatin quality, sperm parameters, and apoptosis following testicular torsion-detorsion in mice

Curcumin mitigates acrylamide‐induced ovarian antioxidant disruption and apoptosis in female Balb/c mice: A comprehensive study on gene and protein expressions

Curcumin and Its Nanoformulations: Exploring Therapeutic Potential in Female Reproductive Health

Reply: Thank you so much for your comments on our manuscript. We added “Curcumin has many benefits for the reproductive system, it has excellent prospects for the treatment of fertility disorders. Curcumin compensates for the deleterious effects of testicular ischaemia and improves sperm chromatin quality in mice[10], mitigates acrylamide-induced ovarian antioxidant disruption and apoptosis in female Balb/c mice[11]” in Background on page 4.

2- Comment: While the rationale for using curcumin is well presented, it might benefit from more information about the gap in current literature that the study aims to address. The introduction could be improved by stating the specific hypotheses or questions being tested more explicitly.

2. Reply: In order to further present the aims of the use of curcumin, we added “Curcumin is a commonly used gynaecological drug globally[5-7].” on page 3.

3- Comment: Please delete the last paragraph in the introduction “In this study we observed elevated oxidative stress, increased DNA damage, elevated mitochondrial reactive oxygen species (ROS) levels, decreased ATP production, increased mitochondrial number, and decreased mitochondrial autophagy in curcumin-treated ovarian cancer cells, indicating that curcumin might exert its anti-tumor effect via inducing mitochondrial stress”, this is the results of your study and it is better to add in the result or discussion.

3. Reply: We deleted the last paragraph in the introduction “In this study we observed elevated oxidative stress, increased DNA damage, elevated mitochondrial reactive oxygen species (ROS) levels, decreased ATP production, increased mitochondrial number, and decreased mitochondrial autophagy in curcumin-treated ovarian cancer cells, indicating that curcumin might exert its anti-tumor effect via inducing mitochondrial stress”, added it in the discussion, and delete previous portion “Our study revealed that curcumin treatment induced oxidative stress and DNA damage, mitochondrial dysfunction to exert its anti-cancer effects.”

4-Comment: The introduction is clear and well-organized but. It is suggested to write the objective of the study as below: whether or not curcumin induces mitochondrial dysfunction has not been defined [10, 13, 14], and the exact processes underpinning curcumin-induced oxidative stress and DNA damage are still to be thoroughly characterized. Our research may provide new insights into how curcumin may exert its anti-tumor effect by inducing mitochondrial stress.

4. Reply: We rewrote the objective of the study as you suggested “whether or not this anticancer property of curcumin is exerted through inducing mitochondrial dysfunction has not been well defined [15, 18, 19], and the exact underpinnings of curcumin-induced oxidative stress and DNA damage are still to be thoroughly characterized. Our research may help clarify whether curcumin exerts its anti-tumor effect by inducing mitochondrial stress.” on page 4.

5-Materials and Methods

It is written in detail and completely.

“Curcumin treatment of cultured cells Curcumin (S1848, Selleck Chemicals) was dissolved in DMSO at a stock con centration of 50 mM and applied to cells in culture at different final concentrations.”

Would you please clarify which cell concentration you used?

5. Reply: We descibed cell concentration “Cells then were seeded in 6-well plates at a density of 0.1×106 cells/ml in complete DMEM medium” in Cell lines and Reagents, and “cell lines (1× 103) were seeded in a 96-well culture plate” in Cell viability assay. We added “Single-cell suspensions were generated for each cell line and 1×103 of cells were seeded into six-well tissue culture plates. Then, cells were exposed to different doses of curcumin. Colonies were scored after 10-14 days. All experiments were repeated at least three times.” in Materials and Methods to describe our experiments more clearly.

6-Results

It is recommended to delete any numbers and percentages in the text of the results that is shown in figures.

6. Reply: We deleted all numbers and percentages in the text of the results that is shown in figures.

7-Discussion

comment: The manuscript does not adequately address the study’s limitations, such as the relevance of in vitro findings to clinical settings or the potential off-target effects of curcumin. A lack of critical reflection can suggest a biased interpretation of the results.

7.Reply: We added “The limitations are objective, such as…” and “relevance of in vitro findings to the clinical setting needs to be further explored”. At the end of Discussion, we have already referred to “bioavailability of curcumin and off-target effects ”.

8-Conclusion:

There is no clear conclusion section. Please add it

8.Reply: We add “Conclusion: Curcumin exerts anti-neoplastic effects by inducing oxidative stress, DNA damage and mitochondrial dysfunction in ovarian cancer cells, offered novel approach to treating ovarian cancer.”

We are grateful for the invaluable feedback provided by the reviewers. Professional advice and scholarly opinions are very important for our manuscript and further research.

With kind regards,

Weipei Zhu, Ph.D.

---

## [Editor Report · Decision Letter 2]

10 Feb 2025

Curcumin induces mitochondrial dysfunction-associated oxidative DNA damage in ovarian cancer cells

PONE-D-24-40005R2

Dear Dr. Weipei Zhu,

We’re pleased to inform you that your manuscript has been judged scientifically suitable for publication and will be formally accepted for publication once it meets all outstanding technical requirements.

Kind regards,

Sanaz Alaeejahromi

Academic Editor

PLOS ONE
---

## [Editor Report · Acceptance letter]

PONE-D-24-40005R2

PLOS ONE

Dear Dr. Zhu,

I'm pleased to inform you that your manuscript has been deemed suitable for publication in PLOS ONE. Congratulations! Your manuscript is now being handed over to our production team.

Kind regards,

on behalf of

Dr. Sanaz Alaeejahromi

Academic Editor

PLOS ONE